Sociality genes are associated with human-directed social behaviour in golden and Labrador retriever dogs

http://orcid.org/0000-0002-6115-7517 Persson Mia E.
Sundman Ann-Sofie
Halldén Lise-Lotte
Trottier Agaia J.
Jensen Per per.jensen@liu.se
AVIAN Behaviour Genomics and Physiology Group, IFM Biology, Linköping University , Linköping , Sweden
Vonk Jennifer
Electronic publication date: 2018 Nov 6
Publication date: 2018
Volume: 6
Electronic Location ID: e5889
Received 2018 Jun 26; Accepted 2018 Oct 8
Copyright: © 2018 Persson et al.
Copyright year: 2018
Copyright holder: Persson et al.
License: This is an open access article distributed under the terms of the Creative Commons Attribution License, which permits unrestricted use, distribution, reproduction and adaptation in any medium and for any purpose provided that it is properly attributed. For attribution, the original author(s), title, publication source (PeerJ) and either DOI or URL of the article must be cited.
License URL: https://creativecommons.org/licenses/by/4.0/

Keywords: Genetics, Dog genetics, Dog behavior, Human-dog communication, Human-directed social behavior, Golden retrievers, Labrador retrievers, Wolf, Domestication, Behavior genetics

Funding: Advanced Research Grant from the European Research Council (ERC), project 322206 “GENEWELL” This project was funded by an Advanced Research Grant from the European Research Council (ERC), project 322206 “GENEWELL.” The funders had no role in study design, data collection and analysis, decision to publish, or preparation of the manuscript.

==============================
Background

Dogs have human-directed social skills that allow them to communicate and cooperate with humans. We have previously identified two loci on chromosome 26 associated with human contact-seeking behaviors during an unsolvable problem task in laboratory beagles (Persson et al., 2016). The aim of the present study was to verify the single nucleotide polymorphisms (SNPs) in additional dog breeds. We also studied how the allele frequencies have changed during domestication and recent selection.

Methods

Dogs of two breeds, 61 golden retrievers and 100 Labrador retrievers, were phenotyped and genotyped, and 19 wolves were genotyped. The Labrador retrievers were divided into common and field type by pedigree data to make it possible to study the effects of recent selection. All dogs were tested in an unsolvable problem task where human-directed social behaviors were scored. DNA from dogs (buccal swabs) and wolves (blood or brain tissue) was analyzed for genotype on two of the previously identified SNP markers, BICF2G630798942 (SNP1) and BICF2S23712114 (SNP2), by pyrosequencing.

Results

There was genetic variation for SNP1 in both dog breeds whereas the wolves were fixed for this polymorphism, and for SNP2 there was variation in both dogs and wolves. For both SNPs, Labrador retriever types differed significantly in allele frequencies. We found associations between SNPs and human-directed social behavior in both dog breeds. In golden retrievers, SNP1 was associated with physical contact variables, for example, with the duration of physical contact with the owner (F2,56 = 4.389, p = 0.017). SNP2 was associated with several behavioral variables in both breeds, among others owner gazing frequency in both golden retrievers (F2,55 = 6.330, p = 0.003) and Labradors (F1,93 = 5.209, p = 0.025).

Discussion

Our results verify the association between the previously identified SNPs and human-directed social behavior scored in an unsolvable problem task. Differences in allele frequencies suggest that these loci have been affected by selection. The results indicate that these genomic regions are involved in human-directed social behavior in not only beagles but in other dog breeds as well. We hypothesize that they may have been important during dog domestication.

Introduction

Social behaviors are complex traits affected by environmental factors as well as by many genes, each with small effects. The genetics of complex traits such as social behavior is difficult to study in humans as it requires a standardized environment, dense genotyping as well as phenotyping of a large number of individuals. The dog (Canis familiaris), on the other hand, has a genome with large haplotype blocks and is thus more convenient for genetic mapping (Lindblad-Toh et al., 2005). In addition, their human-like social skills could make them a suitable model species for human social behavior and disorders.

Through sharing our ecological niche for thousands of years, dogs have developed social talents that, in some cases, have been reported to surpass the skills of our closest relative, the chimpanzee (Pan troglodytes), as well as their own wolf ancestor (Canis lupus) (Hare & Tomasello, 1999, 2005). For example, dogs are able to comprehend human ostensive cues and referential gestures such as pointing and gazing (Lakatos et al., 2012; Soproni et al., 2001). They are also able to communicate with humans through intentional communicative referential gestures involving both attention-seeking and directional-showing behavior (Marshall-Pescini et al., 2013; Miklosi et al., 2000; Passalacqua et al., 2011). Furthermore, dogs have been demonstrated to discriminate between human emotions when viewing facial expressions (Muller et al., 2015). Not even socialized wolves are as prone as dogs to seek human attention (Gacsi et al., 2009; Topal et al., 2005), or to communicate with humans through mutual gazing (Nagasawa et al., 2015). Different hypotheses for how these differences between the wolf and the dog evolved has been proposed. For example, Hare & Tomasello (2005) suggested that selection against fear and aggression toward humans also mediated the evolution of dogs’ social skills. More recently, however, Range & Viranyi (2015) showed that wolves are as attentive to both human and conspecific actions as dogs are. They propose the canine cooperation hypothesis, suggesting that wolf–wolf cooperation established the basis for the evolution of dog–human cooperation.

Human-directed contact-seeking behaviors have specifically been studied using a problem-solving paradigm. Hare & Tomasello (2005) and Miklósi et al. (2003) showed that dogs and socialized wolves differ in their human-directed contact-seeking behavior when faced with an unsolvable task. Whereas wolves were more oriented toward the task, dogs quickly gave up and turned to a nearby human in a help-seeking manner. Hence, behavioral tests involving unsolvable tasks that stimulate communication and attention-seeking toward humans can be used to study dog–human social interactions, including variation between and within breeds (Persson et al., 2015, 2017; Sundman et al., 2018).

When studying dogs’ human-directed social behavior it is important to take into account that dogs form strong attachment bonds with their owners (Topal et al., 1998). It has been suggested that owners function as a secure base influencing persistence in cognitive tasks, because the presence of the owner affects dogs’ duration of task manipulation (Horn, Huber & Range, 2013). Additionally, these authors found that dogs spent more time in proximity of their owner and the presence or absence of the owner affected dogs’ interactions toward the unfamiliar experimenter. It is therefore relevant to analyze social interactions toward an unfamiliar experimenter and the familiar owner separately if both are present during a cognitive task. Consistent with this, we previously found that dogs’ social behaviors directed at owners and those directed at unfamiliar experimenter were separated into different components in a principal component analysis (Sundman et al., 2018).

The dog is not only well-suited for studies on social behavior, the species is also well-suited for studies of the genetics of both simple and complex traits such as social behaviors. The structure of the dog genome with long regions of linkage disequilibrium is particularly suitable for genome-wide studies identifying candidate regions for traits of interest (Lindblad-Toh et al., 2005; Sutter et al., 2004). Regarding human-directed social behavior, heritability estimates show a significant genetic component underlying variation in behaviors as measured in an unsolvable problem task (Persson et al., 2015). Furthermore, genome-wide association studies have identified two candidate regions on chromosome 26 associated with human-directed social behavior in laboratory beagles, with single nucleotide polymorphisms (SNPs) located within the SEZ6L and ARVCF genes (Persson et al., 2016). Interestingly, these genes have previously been associated with human social disorders such as autism for SEZ6L (Chapman et al., 2015) and schizophrenia for ARVCF (Sim et al., 2012). Additionally, Persson et al. (2016) found that the SNP associated with the gene ARVCF is in linkage with three other genes of interest for dogs’ sociability, for example, the COMT gene that has previously been associated with mood regulation in humans (Qayyum et al., 2015).

These earlier findings suggest a possible cross-species genetic basis for social behavior shared between dogs and humans. However, further studies are necessary to verify that the same SNP markers are associated with human-directed contact seeking in other dog breeds. In the present study, we utilized behavioral data collected in two different experiments in two other breeds, each experiment using similar methods for studying dog–human interactions in a standardized behavioral test. In both breeds, we collected DNA and analyzed associations with the previously reported candidate SNPs. The aim of the study was thus to investigate whether there are associations between human-directed social behavior, as measured in an unsolvable problem task, and two candidate SNP markers in groups of golden and Labrador retrievers. It is possible that genetic variants affecting social behavior have been under selection during domestication and more recent breed formations. To provide some tentative data in relation to this, we genotyped wolves and included a dog breed that has undergone recent selection for its cooperative bond with humans, the Labrador retriever. The Labrador retriever has recently been split into a common and a hunting type differing in many aspects of social behavior (Sundman et al., 2016).

Materials and Methods

Ethical note

These studies were carried out in accordance with the relevant guidelines and the ethical permit approved by the regional ethical committee for animal experiments in Linköping, Sweden (permit number: 51-13). All owners had given their informed consent for their dogs’ participation. Wolf samples were donated by the veterinarians at Kolmården Wildlife Park and Borås Animal Park in Sweden. All wolf samples were collected in connection with veterinary motivated procedures and no particular ethical license was therefore required for them.

Subjects

Dogs of the breeds golden and Labrador retrievers were recruited to participate by finding dog owners through social media, local radio and advertisements. In total, 61 golden retrievers (34 females and 27 males) and 100 Labrador retrievers (52 females and 48 males) were tested in the same unsolvable problem task and additionally genotyped for the two candidate SNPs. All dogs were originally recruited for and used in other studies. Golden retrievers participated in a study involving intranasal oxytocin treatment and its effect on behaviors in an unsolvable problem task (Persson et al., 2017) and, due to this, they were required to be at least 4 months of age (mean age 5 ± SE 0.5 years; range 4 months-12 years) and not pregnant or lactating. Labrador retrievers participated in a study on correlations between the unsolvable problem task and other behavior assessments (Sundman et al., 2018) and were required to be at least 1 and not older than 4 years of age (mean age 2.43 ± 0.195). Participating dogs also had to be registered as purebred by the Swedish Kennel Club.

In addition, the same candidate SNPs were genotyped in 21 Scandinavian wolf samples (C. lupus, seven females and 14 males) (Table S1). Out of these, 18 blood samples were donated by Kolmården Wildlife Park, Sweden. These blood samples were old samples collected between 2008 and 2016 upon routine procedures for veterinarian purposes. The wolves at Kolmården Wildlife Park were originally born in captivity and originated from five different animal parks in Scandinavia. Three brain samples, collected after death from reasons unrelated to any scientific studies, were donated by Borås Animal Park, Sweden. These wolves were born in captivity, one in Riga, Latvia, and the other two at the zoo in Borås. For more information on the wolf samples, see Table S1.

By use of pedigree information, Labrador retrievers were divided into two types: common and field. If ancestors for at least three generations back were bred for field work, which can be seen in titles of ancestors, for example, field-trial champion, the dogs were classified as field-type Labrador retrievers. If ancestors instead had show titles, they were classified as common-type Labrador retrievers. Labrador retrievers with mixed ancestry were not included in the study. We used the Swedish Kennel Club’s online registry (Hunddata, http://hundar.skk.se/hunddata/) and k9data.com (http://www.k9data.com/) for pedigrees. Based on the pedigree analysis we classified 52 Labrador retrievers as common (28 females and 24 males) and 48 as field type (24 females and 24 males). For golden retrievers, we did not have sufficient information to perform a similar division, and they were therefore all treated as one single breed.

Procedure

Upon arrival at the testing site, owners were informed of the testing procedure. Buccal DNA samples were collected either prior to the testing (golden retrievers) or after the testing (Labrador retrievers). To assure sufficient food motivation in the dogs, their willingness to eat the treats used in the unsolvable problem task was confirmed as described in Persson et al. (2015). Briefly, dogs were presented with three quarter-pieces of Frolic© on a plastic plate of the same material as the problem-solving device but without a lid. The treats were presented one at the time. When the dog ate a treat, another was placed until the dog had consumed the three treats. All subjects consumed all pieces within 20 s and were therefore not considered to differ in their willingness to eat the treats.

Two female experimenters tested the dogs, one person tested the golden retrievers and one the Labrador retrievers. The video analyses were performed by the two experimenters.

Subjects of the breed golden retriever were part of a parallel study investigating effects of oxytocin treatment on dogs’ human directed social behavior (Persson et al., 2017). Therefore, as part of this parallel study, subsequently to DNA sampling half of the females and half of the males received an intranasal dose of 20 IU oxytocin 45 min prior to the behavioral test. Individuals that were not given an oxytocin treatment were instead given saline as a control treatment. After the food motivation test, these dogs were taken for a 30-min walk followed by 10 min of resting in the car immediately prior to the behavior test. Treatment was taken into account in the later analyses, as described below. The experimenter who tested the golden retrievers was blinded to which hormone treatment the dogs received until after the behavioral video analysis. Subjects of the breed Labrador retriever had been subjected to a standardized test battery (Behavior and Personality test for Dogs, Swedish Kennel Club) before testing them in the unsolvable problem task. After that, they were tested in a pointing test.

Unsolvable problem task

Testing was carried out at ten different locations in Sweden during the autumn of 2014 (Labrador retrievers) and autumn of 2015 (golden retrievers). Results from the behavior test have been previously published in Sundman et al. (2018) (Labrador retrievers) and in Persson et al. (2017) (golden retrievers). The unsolvable problem task was performed in the same way in both breeds. To have a uniform setting, testing took place in a 3 × 3 m marquee tent without the presence of any other dogs. The tent had three walls and no flooring. A mesh fence was placed at the open side to keep the dogs within the tent (testing area). An HD camcorder (Canon Legria HF G25) was placed on a camera stand approximately three m from the testing area to record the behavior of each dog.

Dogs were tested with the unsolvable problem task thoroughly described in Persson et al. (2015) with the addition of the presence of the owner. The device used consists of a plastic tray (55 × 25 cm) with three identical circular wells (seven cm in diameters) covered with plexiglass lids with odor ports (Fig. 1). Three quarter-pieces of Frolic© dog treats were placed underneath each lid. The dogs could easily access the treats in two of the three wells by sliding the lids to the side. However, the lid in the middle could not be opened hence making the task unsolvable. The experimenter cleaned, prepared and placed the unsolvable problem task on the ground approximately 15–30 cm from the middle of the back wall prior to the arrival of each dog.

Figure 1 The unsolvable task.

(A) Golden retriever dog interacting with the unsolvable task. (B) The plastic tray measures 55 × 25 cm, circular wells seven cm in diameter and the plexiglass lids 10 × 15 cm. The left and right lid can be opened to access the treats. The middle lid cannot be opened, hence making the task unsolvable. Photo credit: Mia E. Persson.

Upon arrival at the testing arena, owners were reminded to stand passively immediately close to the fence at the front right corner inside the tent, facing the problem task. The experimenter was standing in the same position but on the opposite side of the tent (in the front left corner). After closing the fence gate, the owner was asked to unleash the dog that then could freely move around inside the testing area from this point onward. The owner had been instructed to not interact with the dog unless it was attempting to escape. If the dog tried to leave the tent, the owner was allowed to interrupt and call the dog back. Behaviors were not recorded during this interruption. If the dog had not opened any of the lids within 60 s, the experimenter opened both solvable lids halfway and immediately went back to her original position. The duration of the behavior test was 3 min.

The behaviors scored from the behavior test were the human-directed social behaviors proximity, physical contact, and gazing in relation to owner and experimenter. Duration and frequency were scored for each of these behaviors as described in the ethogram (Table 1). In addition to the social behaviors, we scored the duration of the time the dog spent in close proximity of the test-setup. The behaviors were scored from the video recordings using the Observer XT 10, Noldus software (https://www.noldus.com/knowledge-base/observer-xt-10).

Table 1 Ethogram of the behaviors analyzed in the unsolvable problem task.

Behavior	Description	
Experimenter zone	Dog’s head is within one body length of the experimenter	
Owner zone	Dog’s head is within one body length of the owner	
Experimenter gaze	The dog directs its eyes towards the face of the experimenter	
Owner gaze	The dog directs its eyes towards the face of the owner	
Experimenter physical contact	The dog is in physical contact with the experimenter	
Owner physical contact	The dog is in physical contact with the owner	
Duration test-setup	The duration of time (s) the dog’s head is within one body length of the test-setup	
Note:

Duration and frequency of the behaviours were scored. Zone behaviours were mutually exclusive.

DNA sampling, extraction, and genotyping

Buccal cells were collected from the dogs by rubbing a cotton swab on the inside of their cheek for approximately 20 s. Buccal samples were stored at 4 °C and wolf samples (blood and serum) were stored at −20 °C until DNA extraction. The standard protocol of the Isohelix DDK-50 kit was used to extract DNA from buccal swabs, with the exception that samples were kept in Lysis Buffer and proteinase K for 48 h prior to continuing with the protocol. Single 50 μl elusions were used. DNA was also extracted from fifteen whole blood and three serum wolf samples using the QIAGEN DNeasy® Blood and Tissue Kit and from three wolf brain samples using QIAGEN AllPrep DNA/RNA/miRNA Universal Kit, both by standard protocol. Subsequently, DNA yield was quantified using a Nanodrop ND-1000 and all isolated samples were stored at −20 °C until further use.

Genotyping was performed on the two SNPs identified in Persson et al. (2016), BICF2G630798942 (rs23313128, chr26:20025266C/A) and BICF2S23712114 (rs23317526, chr26:29319675A/G). Hereafter, BICF2G630798942 will be referred to as SNP1 and BICF2S23712114 as SNP2. Polymerase chain reaction (PCR) and subsequent pyrosequencing were used to genotype both wolf and dog samples for the two SNPs. Primers were designed using the PyroMark Assay Design software by QIAGEN©. The primers used for SNP1 were: forward biotinylated in 5′ CTGCCAGGGACTCCTGAG, reverse CTCAAGGCAGCCCATCACT and sequencing reverse GGAGGCTTGCTGCCG. For SNP2 the primers used were: forward biotinylated in 5′ CATGTCACAGTTGAGGGGATAGGT, reverse TCTTCAGACAGCCCACCCA and sequencing reverse CAGTCCAGGAAGGAATA. For each sample, the PCR-mixture contained 0.12 μl DreamTaq™ DNA Polymerase 5 u/μl (Thermo Scientific, Waltham, MA, USA), 2.5 μl of 10X DreamTaq™ Buffer (Thermo Scientific, Waltham, MA, USA), 0.5 μl dNTP 10 mM (2.5 mM each, BIOLINE), 0.5 μl of each primer diluted to 5 μM (Invitrogen, Carlsbad, CA, USA), 19.9 μl of nuclease free water and approximately 100–200 ng of DNA template. The final PCR volume was 25 μl for each sample and the reaction was run on the Palmcycler PCR by Corbett. The PCR cycle consisted of an initial denaturation at 95 °C for 3 min, 40 cycles of 30 s denaturation at 95 °C, 30 s annealing at 63 °C for the SNP1 primers and 61 °C for SNP2 primers, 30 s extension at 72 °C and a 10 min final extension at 72 °C. Pyrosequencing was performed on the entire PCR product according to the PyroMark Q24 Vacuum Workstation Quick-Start Guide found at www.qiagen.com. The results were analyzed using the PyroMark Q24 2.0.6 software.

Genotyping of the SNP1 marker was successful in all 61 golden retriever samples, in 97 Labrador retriever samples (genotyping failed in one male and two female samples) and in 19 wolf samples (two female samples failed). SNP2 was successfully genotyped in 60 golden retrievers (one female sample failed), in 98 Labrador retrievers (two female samples failed) and in 19 wolves (two female samples failed).

Statistical analysis

Except for Hardy–Weinberg estimates (HWE), all statistical analyses were carried out using IBM SPSS statistics software version 22 and 25. Behavior data was checked for normality both visually and with the Kolmogorov–Smirnov test, and, if necessary, transformed using log10 (x+1). To analyze behavior data, generalized linear mixed models were used. For golden retriever analyses, statistical models contained oxytocin treatment, sex and SNP genotype and for Labrador retriever analyses, the models contained type, sex and SNP genotype. The two SNPs were tested separately together with the other fixed variables and Bonferroni correction was used to account for multiple testing in post hoc comparisons. Data distribution was set to normal with a link function or gamma with a log function depending on data distribution. Best model-distribution fit was determined by Akaike measurements comparisons. Final models for each behavior including F and p statistics can be found in Table 2 for golden retrievers and Table 3 for Labrador retrievers. Inter-observer reliability analysis was done for all behaviors in 10% of the individuals with correlation coefficients ranging from 0.901 to 0.999 (Pearson, London, UK) and 0.803 to 1 (Spearman, London, UK) (Table S2). Fisher’s exact test was used to compare genotype frequencies. To calculate HWE, the exact test incorporated in the “genetics” R-package was used.

Table 2 Generalized linear mixed models analysis for behavioral variables and SNP genotype in golden retrievers.

Behavior	Df1	Df2	SNP1	Sex	Treatment	SNP1 * sex	SNP1 * treatment	Distribution	
F	p	F	p	F	p	F	p	F	p	
Duration experimenter look	1	56	1.723	0.188	7.382	0.009	0.058	0.81		NA		NA	Gamma log	
Frequency experimenter look	1	56	0.575	0.566	1.537	0.22	0.615	0.436		NA		NA	Gamma log	
Duration experimenter zone	1	56	0.096	0.908	1.742	0.192	0.488	0.488		NA		NA	Normal identity	
Frequency experimenter zone	1	56	0.292	0.748	1.112	0.296	0.085	0.771		NA		NA	Gamma log	
Duration experimenter contact	1	56	2.117	0.13	0.569	0.454	15.797	>0.001		NA		NA	Gamma log	
Frequency experimenter contact	1	56	4.339	0.018	0.697	0.407	9.705	0.003		NA		NA	Gamma log	
Duration owner look	1	56	1.68	0.196	0.006	0.938	0.274	0.603		NA		NA	Gamma log	
Frequency owner look	2	56	1.26	0.292	0.002	0.965	2.356	0.13		NA		NA	Gamma log	
Duration owner zone	1	56	0.451	0.639	6.59	0.013	1.194	0.279		NA		NA	Gamma log	
Frequency owner zone	1	56	1.956	0.151	2.197	0.144	0.292	0.663		NA		NA	Gamma log	
Duration owner contact	2	56	4.389	0.017	1.095	0.3	2.93	0.092		NA		NA	Gamma log	
Frequency owner contact	2	56	6.996	0.002	3.175	0.08	8.137	0.006		NA		NA	Gamma log	
Duration test-setup	2	56	0.457	0.636	3.246	0.077	0.217	0.643		NA		NA	Gamma log	
Behavior	Df1	Df2	SNP2	Sex	Treatment	SNP2 * sex	SNP2 * treatment	Distribution	
F	P	F	P	F	P	F	P	F	P	
Duration experimenter look	1	55	1.023	0.366	7.682	0.008	0.013	0.909		NA		NA	Gamma log	
Frequency experimenter look	2	55	3.243	0.047	2.957	0.091	0.771	0.384		NA		NA	Gamma log	
Duration experimenter zone	2	55	3.304	0.044	1.655	0.204	0.552	0.461		NA		NA	Normal identity	
Frequency experimenter zone	1	55	0.97	0.386	1.185	0.281	0.051	0.822		NA		NA	Gamma log	
Duration experimenter contact	1	55	1.113	0.336	0.099	0.754	22.377	>0.001		NA		NA	Gamma log	
Frequency experimenter contact	1	55	2.711	0.075	0.125	0.725	12.305	0.001		NA		NA	Gamma log	
Duration owner look	1	55	4.477	0.016	0.319	0.575	>0.001	1		NA		NA	Gamma log	
Frequency owner look	2	55	6.33	0.003	0.129	0.72	3.658	0.061		NA		NA	Gamma log	
Duration owner zone	1	55	0.14	0.87	8.106	0.006	2.42	0.126		NA		NA	Gamma log	
Frequency owner zone	1	55	1.536	0.224	2.347	0.131	1.529	0.222		NA		NA	Gamma log	
Duration owner contact	2	55	14.809	>0.001	3.802	0.056	5.438	0.023		NA		NA	Gamma log	
Frequency owner contact	1	55	0.028	0.972	4.285	0.043	3.318	0.074		NA		NA	Gamma log	
Duration test-setup	2	55	0.099	0.906	2.882	0.095	0.483	0.490		NA		NA	Gamma log	
Notes:

The models also included sex and treatment (intranasal oxytocin) as fixed factor.

Significant (P < 0.05) results are shown in bold.

Table 3 Generalized linear mixed models analysis for behavioral variables and SNP genotype in Labrador retrievers.

Behavior	Df1	Df2	SNP1	Sex	Type	SNP1 * sex	SNP1 * type	Distribution	
F	P	F	P	F	P	F	P	F	P	
Duration experimenter look	2	91	2.472	0.09	0.043	0.837	10.475	0.002		NA	2.912	0.091	Gamma log	
Frequency experimenter look	2	91	1.719	0.185	0.148	0.702	7.679	0.007		NA	2.962	0.089	Gamma log	
Duration experimenter zone	1	92	1.031	0.351	0.113	0.737	0.006	0.941		NA		NA	Gamma log	
Frequency experimenter zone	1	92	1.09	0.34	0.698	0.406	9.046	0.003		NA		NA	Gamma log	
Duration experimenter contact	1	92	1.582	0.211	0.047	0.828	0.046	0.83		NA		NA	Gamma log	
Frequency experimenter contact	1	92	1.085	0.342	0.131	0.719	0.274	0.602		NA		NA	Gamma log	
Duration owner look	2	90	0.953	0.39	8.82	0.004	27.878	>0.001		NA	10.394	0.002	Gamma log	
Frequency owner look	2	91	1.242	0.294	2.897	0.092	20.208	>0.001		NA	2.711	0.103	Gamma log	
Duration owner zone	1	92	0.026	0.975	0.085	0.772	3.147	0.079		NA		NA	Normal identity (logged)	
Frequency owner zone	1	92	1.432	0.244	0.029	0.866	18.153	>0.001		NA		NA	Gamma log	
Duration owner contact	1	90	0.034	0.966	0.211	0.647	15.91	>0.001		NA		NA	Gamma log	
Frequency owner contact	1	92	0.268	0.765	0.177	0.675	8.065	0.006		NA		NA	Gamma log	
Duration test-setup	2	92	0.020	0.980	0.211	0.647	1.946	0.166		NA		NA	Gamma log	
Behavior	Df1	Df2	SNP2	Sex	Type	SNP2 * sex	SNP2 * type	Distribution	
F	P	F	P	F	P	F	P	F	P	
Duration experimenter look	2	93	3.365	0.07	0.087	0.769	0.056	0.813		NA	7.796	0.006	Gamma log	
Frequency experimenter look	2	93	3.025	0.085	0	0.992	0.122	0.728		NA	2.314	0.132	Gamma log	
Duration experimenter zone	1	94	6.252	0.014	0.312	0.578	0.135	0.714		NA		NA	Gamma log	
Frequency experimenter zone	1	94	6.86	0.01	0.643	0.425	4.821	0.031		NA		NA	Gamma log	
Duration experimenter contact	1	94	3.316	0.072	0.283	0.596	0.462	0.498		NA		NA	Gamma log	
Frequency experimenter contact	1	94	3.594	0.061	0.398	0.53	0.348	0.557		NA		NA	Gamma log	
Duration owner look	2	93	7.209	0.009	13.995	>0.001	25.596	>0.001	3.509	0.064		NA	Gamma log	
Frequency owner look	2	93	5.209	0.025	1.729	0.192	1.807	0.182		NA	3.947	0.05	Gamma log	
Duration owner zone	1	94	5.945	0.017	0.302	0.584	2.387	0.126		NA		NA	Normal identity (logged)	
Frequency owner zone	1	94	13.846	>0.001	0.104	0.747	10.559	0.002		NA		NA	Gamma log	
Duration owner contact	1	94	1.48	0.227	0.068	0.794	20.862	>0.001		NA		NA	Gamma log	
Frequency owner contact	1	94	2.879	0.093	0.043	0.836	9.025	0.003		NA		NA	Gamma log	
Duration test-setup	1	94	2.942	0.090	0.479	0.491	2.141	0.147		NA		NA	Gamma log	
Notes:

The models also included sex and type (common or field) as fixed factors.

Significant (P < 0.05) results are shown in bold.

Results

Variation was found in SNP1 in golden retrievers (HWE: p = 0.769) and in Labrador retrievers (HWE: p = 0.153) but all wolves were fixed for the C allele (Fig. 2A). For SNP2, variation was found in both golden retrievers (HWE: p = 1), in Labrador retrievers (HWE: p = 1) and in wolves (HWE: p = 1) (Fig. 2B). Additionally, when looking at the two types of Labrador retriever separately, there was a variation in both SNPs for both the common and the field type and the HWE was not significant for any of them (for SNP1 p = 0.580 and p = 1 and for SNP2 p = 1 and p = 1 for common and field, respectively) (Fig. 2).

Figure 2 Allele frequencies.

Allele frequencies for (A) the SNP BICF2G630798942 (SNP1) and (B) the SNP BICF2S23712114 (SNP2) for Labrador retrievers (common and field type), golden retrievers, and wolves.

The Fisher’s exact test showed a significant difference in genotype frequencies between the common and field Labrador retrievers for both SNP1 (p < 0.001) and SNP2 (p = 0.031) (Fig. 2). For SNP1, AC was the most common genotype in common type Labrador retrievers whereas CC was the most common in the field type. In SNP2, AA was the most frequent genotype for both types, but the proportion between AA and AG differed between the types. In the common type a larger proportion was of the AG genotype than in the field type.

In the unsolvable problem task, there were no associations between the SNPs and the time spent close to the test-setup. However, we found effects of both SNPs on behavioral variables related to social behavior in both breeds. The final models for each social behavior variable as well as F and p statistics for all behaviors can be found in Tables 2 and 3. Figures of all genotype-behavior associations are presented in Fig. S1. In the golden retriever, the genotype of SNP1 was significantly associated with physical contact, both on the frequency with the experimenter (Fig. 3A, F1, 56 = 4.339, p = 0.018) and on frequency (Fig. 3B, F2, 56 = 6.996, p = 0.002) and duration (Fig. 3C, F2, 56 = 4.389, p = 0.017) with the owner. Golden retrievers with the AA genotype had a higher frequency of physical contact with the experimenter than individuals with AC. The AC genotype had instead more frequent physical contact with their owners than both AA and CC, and genotypes AA and AC had contact for a longer duration than individuals with the CC genotype.

Figure 3 Associations between SNP1 and behaviors in golden retrievers.

Associations between BICF2G630798942 (SNP1) and the frequency/duration for behaviors scored in the unsolvable problem task in golden retrievers. There was a significant difference between the genotypes in (A) the frequency of physical contact with the experimenter, and (B) and (C) the frequency and duration of physical contact with the owner. Graphs show mean frequency/duration for each of the genotypes. Error bars show ± 1 SE. *p < 0.05.

In the Labrador retrievers, we found a significant interaction between genotype of SNP1 and breed type for the duration of gazing at owner (Fig. 4, F1, 90 = 10.394, p = 0.002). In the field type, dogs with CC genotype gazed longer at their owner than those with AC genotype, whereas there were no differences between the genotypes among the common type.

Figure 4 Associations between SNP1 and behaviors in Labrador retrievers.

For the duration of owner gaze there was an interaction between genotype and type. The SNP BICF2G630798942 (SNP1) was associated with owner gaze in the field type but not in the common type. Graph shows mean duration for each of the genotypes. Error bars show ± 1 SE. *p < 0.05.

SNP2 was significantly associated with several human-directed social behaviors in both breeds. In the golden retrievers, AG individuals spent less time in the experimenter zone than AA and GG dogs (Fig. 5A, F2, 55 = 3.304, p = 0.044) and AA dogs gazed at the experimenter more frequently than GG dogs (Fig. 5B, F2, 55 = 3.243, p = 0.047). Individuals carrying the AA and AG genotype gazed at their owners more frequently (Fig. 5C, F2, 55 = 6.330, p = 0.003) and with a longer duration (Fig. 5D, F1, 55 = 4.477, p = 0.016) than GG dogs. Also, golden retrievers with the AG genotype had longer duration of physical contact with their owner than those with AA (Fig. 5E, F2, 55 = 14.809, p < 0.001).

Figure 5 Associations between SNP2 and behaviors in golden retrievers.

Associations between the genotype on the SNP BICF2S23712114 (SNP2) and the frequency/duration for behaviors scored in the unsolvable problem task in golden retrievers. There was a significant difference between the genotypes in (A) the duration in the experimenter zone, (B) the frequency of gazing at the experimenter, (C) and (D) the frequency and duration of gazing at the owner, and (E) the duration of physical contact with the owner. Graphs show mean frequency/duration for each of the genotypes. Error bars show ± 1 SE. *p < 0.05.

In the Labrador retrievers, AA individuals at SNP2 spent more time in the experimenter zone (Fig. 6A, F1, 94 = 6.252, p = 0.014) and visited it more frequently (Fig. 6B, F1, 94 = 6.860, p = 0.010), as well as spent more time in the owner zone (Fig. 6C, F1, 94 = 5.945, p = 0.017) and visited it more frequently (Fig. 6D, F1, 94 = 13.846, p < 0.001). AA individuals also gazed more often at their owner (Fig. 6E, F1, 93 = 5.209, p = 0.025) whereas AG individuals instead gazed at their owner for a longer duration (Fig. 6F, F1, 93 = 7.209, p = 0.009). For the duration of gazing towards experimenter (Fig. 7A) and the frequency of gazing towards the owner (Fig. 7B) there was an interaction between genotype and type in the Labrador retrievers (F1, 93 = 7.796, p = 0.006 and F1, 93 = 3.947, p = 0.05, respectively). In the field type, AA individuals gazed longer at the experimenter and more often at their owner than AG individuals whereas there were no differences in the common type.

Figure 6 Associations between SNP2 and behaviors in Labrador retrievers.

Associations between the genotype on the SNP BICF2S23712114 (SNP2) and the frequency/duration for behaviors scored in the unsolvable problem task in Labrador retrievers. There was a significant difference between the genotypes in (A) and (B) the duration and frequency in the experimenter zone, (C) and (D) the duration and frequency in the owner zone, and (E) and (F) the frequency and duration of gazing at the owner. Graphs show mean frequency/duration for each of the genotypes. Error bars show ± 1 SE. *p < 0.05.

Figure 7 Interactions between SNP2 and type in Labrador retrievers.

For (A) the duration of experimenter gaze, and (B) the frequency of owner gaze, there was an interaction between the SNP BICF2S23712114 (SNP2) and type (common or field) in Labrador retrievers. While genotype was associated with both behaviors in the field type Labradors, they were not associated with the behaviors in the common type. Graphs show mean frequency/duration for each of the genotypes. Error bars show ± 1 SE. *p < 0.05.

In golden retrievers, sex and intranasal oxytocin treatment (part of the experiment for which the dogs were originally recruited) were included in the models. In the analyses for both SNP1 and SNP2, sex had an effect on the duration dogs spent in the owner zone where males spent significantly more time with their owner than females (F1, 56 = 6.590, p = 0.013 for SNP1 model and F1, 55 = 7.374, p = 0.009 for SNP2 model; males 35.02 ± 7.986 vs. females 16.72 ± 3.921). Additionally, in analyses for both SNPs, intranasal oxytocin treatment significantly decreased the duration (F1, 56 = 15.797, p < 0.001 and F1, 55 = 22.377, p < 0.001; oxytocin 0.24 ± 0.109 vs. control 1.75 ± 0.591) and frequency (F1, 56 = 9.705, p = 0.003 and F1, 55 = 12.305, p = 0.001; oxytocin 0.97 ± 0.481 vs. control 4.20 ± 1.603) of experimenter physical contact seeking as well as the frequency of owner physical contact (F1, 56 = 8.137, p = 0.006 and F1, 55 = 5.437, p = 0.023; oxytocin 0.71 ± 0.377 vs. control 1.67 ± 0.615).

For the Labrador retrievers, sex and type were included in the models. In the models for both SNP1 and SNP2, the difference between males and females for the duration of gazing at their owner was significant where males looked for a longer time (F1, 90 = 7.667, p = 0.004 and F1, 93 = 13.995, p < 0.001; 10.19 ± 2.72 vs. 4.42 ± 0.73 s for males and females, respectively). There were many differences between the types. In the models for both SNPs, the frequency of owner and experimenter zone were significant as well as the duration of gazing at the owner and frequency and duration of owner physical contact. Field type Labradors visited the experimenter zone at a higher frequency (F1, 92 = 9.046, p = 0.003 and F1, 94 = 4.821, p = 0.031; 3.27 ± 0.37 vs. 2.12 ± 0.22) as well as the owner zone (F1, 92 = 18.153, p < 0.001 and F1, 94 = 10.559, p = 0.002; 4.21 ± 0.43 vs. 2.51 ± 0.23 times). The field type gazed longer at their owner (F1, 91 = 10.475, p = 0.002 and F1, 93 = 25.596, p < 0.001; 5.74 ± 0.92 vs. 3.16 ± 0.50 s) and were in physical contact with their owner both more often (F1, 92 = 8.065, p = 0.006 and F1, 94 = 20.862, p < 0.001; 0.77 ± 0.17 vs. 0.29 ± 0.10) and for a longer time (F1, 90 = 15.910, p < 0.001 and F1, 94 = 9.025, p = 0.003; 0.70 ± 0.19 vs. 0.09 ± 0.04). Additionally, in the model for SNP1, there were significant differences between the types in frequency and duration of gazing at the experimenter as well as frequency of owner gazing. The field type gazed both longer and more often at the experimenter (F1, 91 = 10.475, p = 0.002 and F1, 91 = 7.679, p = 0.007) and more often at their owner (F1, 91 = 20.208, p > 0.001). The means of the duration and frequency for looking at experimenter were 5.74 ± 0.92 s and 4.02 ± 0.49 times in field type vs. 3.16 ± 0.50 s and 2.63 ± 0.34 in the common type. For the frequency of looking at owner, the means were 6.52 ± 0.82 vs. 3.00 ± 0.33 times.

Discussion

In a previous genome-wide association study on beagles tested in the unsolvable problem paradigm, Persson et al. (2016) identified two SNPs on chromosome 26, BICF2G630798942 (SNP1) and BICF2S23712114 (SNP2), associated with social interactions directed toward humans. Here, we show that both SNPs are also associated with human-directed social behavior in two additional dog breeds, golden and Labrador retriever. We also show that genotype frequencies for the two SNPs differ between wolves and dogs, between breeds and between recently selected breed types. Thus, these loci could have been affected by selection during domestication as well as during breed formations. This suggests that selection of favorable alleles in the genomic region of the SNPs may have been an important part of dog domestication.

In a population of laboratory beagles, Persson et al. (2016) found a significant association between SNP1 and the duration of physical contact and duration of human proximity. Additionally, a suggestive association was found between SNP2 and the duration of human proximity. It is important to stress that these SNPs are not causative for the behavior differences but rather linked to the specific causal loci. SNP1 and SNP2 are both located on chromosome 26, and within the linkage disequilibrium regions five possible associated genes are present. The marker SNP1 is located in an intron of the gene SEZ6L and there are no other genes present in the same linkage block. SNP2 is located in an intron of the gene ARVCF and three additional genes are in linkage: COMT, TXNRD2, and TANGO2. Previously, SEZ6L, ARVCF, COMT, and TXNRD2 have been associated with social disorders and schizophrenia in humans (Chapman et al., 2015; Mas et al., 2010; Sanders et al., 2005; Xu et al., 2013).

In the present study, we verify that the association between these genomic regions and human-directed social behavior are not specific to the previously studied beagles. Similar effects were found in two other breeds, Labrador and golden retrievers, tested in the same unsolvable problem task as the beagles. The genotype of SNP1 had an effect on physical contact seeking with both experimenter and owner in the golden retriever and was associated with a difference in owner gazing between Labrador types. The genotype of SNP2 was primarily associated with eye contact seeking with the experimenter as well as owner in both golden and Labrador retrievers. In the Labradors this effect differed between types where only field type individuals carrying the AA-genotype were gazing more. Additionally, SNP2 was associated with experimenter proximity seeking in both breeds and owner proximity seeking in only Labradors. Finally, an association was found between SNP2 and owner physical contact seeking in golden retrievers but not in Labradors. Genotype and sex interactions were identified in the beagles for SNP1 (Persson et al., 2016) but this was not present in the retrievers.

In addition to SEZ6L, ARVCF, and COMT, there are other genes suggested to be associated with dogs’ human-directed social behavior. Previous research has found associations between polymorphisms in the oxytocin receptor gene and social behavior both during problem solving and in additional situations (Kis et al., 2014; Persson et al., 2017). The oxytocin receptor gene has also been associated with successful training of detection dogs (Konno et al., 2018). Another gene of interest is the dopamine receptor D4 gene that has been associated with gazing toward humans (Hori et al., 2013). Additionally, VonHoldt et al. (2017) investigated a candidate region associated to Williams–Beuren syndrome (WBS) in humans, which is causing hyper-social behavior amongst other effects. It was found that structural variants in this region were also associated with extreme sociability in dogs.

During the course of domestication, dogs seem to have evolved impressive interspecific cooperation skills with humans (Jensen et al., 2016; Miklósi & Topál, 2013). For example, comparative studies have shown that dogs have a higher sociability in general than wolves (Bentosela et al., 2016) and, specifically, that dogs seek more human contact when faced with a problem than wolves do (Heberlein et al., 2016; Miklósi et al., 2003; Udell, 2015). It seems that genetic variants contributing to human-directed social abilities have been selected during domestication and, thus, the two genomic regions investigated in the present study may have been targeted. Variation in sociability toward humans has a significant genetic component, which has been shown by heritability estimates (Persson et al., 2015; Sundman et al., 2016; Van Der Waaij, Wilsson & Strandberg, 2008; Wilsson & Sundgren, 1998) and this is a requirement for selection. Wolves were found to be fixed for the C-allele on SNP1 whereas there was a variation in SNP2. The former polymorphism may thus not exist or is rare among wolves and could even have appeared after the split with dogs, whereas the latter is present in both species. However, as this current study has investigated the SNP genotypes in only a very limited sample of Scandinavian wolves, these results should be cautiously interpreted. A recent study describing the genetic architecture of a dog x wolf crossbreed, the Czechoslovakian Wolfdog, found regions containing excess of wolf and dog ancestry genes respectively (Caniglia et al., 2018). The Czechoslovakian Wolfdog derives from a cross between Carpathian wolves and German shepherd dogs and, even though it shares many morphological features with the wolf, it shows mostly dog-like behavioral phenotypes. Interestingly, the SEZ6L, ARVCF, and COMT genes were all detected within regions of excess dog ancestry.

When using an unsolvable problem paradigm to study human-directed social behavior we should also consider the dogs’ persistence to solve the problem and their food motivation. Previous studies have, for example, found a negative correlation between persistence and eye-contact duration (Brubaker et al., 2017), and that more persistent dogs show a longer latency until they gaze at the present human (Marshall-Pescini et al., 2017). On the other hand, in both Persson et al. (2015) and Sundman et al. (2018), test-setup interactions form a component separate from social behaviors in a principal component analysis. It is difficult to disentangle sociability from persistence and food motivation when using the unsolvable problem paradigm. However, neither of the two SNPs in the present study were associated with the duration the dogs spent in proximity of the test-setup, and thus, they seem to be associated with human-directed social behavior rather than persistence.

Previous studies have reported breed differences in behaviors related to human communication (Passalacqua et al., 2011; Sundman et al., 2018; Udell et al., 2014). We can study recent selection by examining established breeds recently diverged into two types due to different breeding goals; for example, dogs suitable for field-work vs. pet and conformation dogs. The Labrador retriever is a breed clearly divided into two types. Pedigrees as well as morphology and behavior distinguish the common type from the field type (Sundman et al., 2016, 2018). Specifically, Sundman et al. (2018) compared the types in the unsolvable problem task and found several behavioral differences. During recent selection, it seems that the field type has increased its human-contact seeking behavior in comparison to the common type, although environmental causes cannot be discarded. Likewise, in the present study, we found differences in allele frequencies between the types for both SNP1 and SNP2. These genetic markers have thus been affected by recent selection in two selection lines of dogs that also differ in social abilities, lending further support to the association between the genomic region and human-directed social behavior.

Conclusions

Our results verify the associations between human-directed social behavior and the SNPs BICF2G630798942 and BICF2S23712114 on canine chromosome 26. We suggest that these loci could have been affected by domestication and selection for sociality in dogs and that genetic variants linked to the SNPs may have been targeted during domestication. Hence, genes within the linkage disequilibrium of these genetic markers are of high interest for further investigation of the genetics behind the impressive social skills of dogs.

Supplemental Information

Supplemental Information 1 Detailed information about the Scandinavian wolf samples.

Click here for additional data file.

Supplemental Information 2 Inter-observer correlations of 10% of the individuals coded with the unsolvable task.

In cases of few observations of a coded behaviour the correlation values are replaced with “NA”.

Click here for additional data file.

Supplemental Information 3 Figures of all genotype-behaviour associations for both golden and Labrador retrievers.

Page 2 shows SNP1 and experimenter-directed interactions; page 3 shows SNP1 and owner-directed interactions; page 4 shows SNP2 and experimenter-directed interactions; page 5 shows SNP2 and owner-directed interactions.

Click here for additional data file.

Supplemental Information 4 Raw behavioural and genotype data from golden and Labrador retrievers tested with the unsolvable task.

Click here for additional data file.

We wish to thank the enthusiastic dog owners who allowed their dogs to participate in this project as well as Borås Zoo and Kolmården Wildlife Park for kindly providing us with wolf samples. Special thanks to Therese Hård at Borås Zoo for all your help, and to Lina S.V. Roth and Enya van Poucke, Linköping University, for support and for proofreading of the manuscript.

Additional Information and Declarations

Competing Interests

Author Contributions

Animal Ethics

Data Availability

The authors declare that they have no competing interests.

Mia E. Persson conceived and designed the experiments, performed the experiments, analyzed the data, contributed reagents/materials/analysis tools, prepared figures and/or tables, authored or reviewed drafts of the paper, approved the final draft.

Ann-Sofie Sundman conceived and designed the experiments, performed the experiments, analyzed the data, contributed reagents/materials/analysis tools, prepared figures and/or tables, authored or reviewed drafts of the paper, approved the final draft.

Lise-Lotte Halldén performed the experiments, approved the final draft.

Agaia J. Trottier performed the experiments, approved the final draft.

Per Jensen conceived and designed the experiments, contributed reagents/materials/analysis tools, authored or reviewed drafts of the paper, approved the final draft.

The following information was supplied relating to ethical approvals (i.e., approving body and any reference numbers):

These studies were carried out in accordance with the relevant guidelines and the ethical permit approved by the regional ethical committee for animal experiments in Linköping, Sweden (permit number: 51-13). All owners had given their informed consent for their dogs’ participation.

The following information was supplied regarding data availability:

Raw data are provided in the Supplemental Files.

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
