# Peer review of "Sociality genes are associated with human-directed social behaviour in golden and Labrador retriever dogs"

_PeerJ, doi:10.7717/peerj.5889_

## Round 0.1 · original submission · Major Revisions

I was extremely fortunate to receive two very thorough and helpful reviews from experts in this area. Both reviewers made very clear and reasonable suggestions for ways to improve your MS and I am in complete agreement with their suggestions. Given the importance of your topic, and the interest that I am sure your work will generate, I would like to invite you to revise your MS. However, the suggested revisions will be extensive and so I have not done the usual line by line editing at this stage. However, I do have a few general comments of my own to add to the reviewers’ detailed comments.

The rationale for the study design is not fully fleshed out. This could be partially due to the fact that, as Reviewer 2 suspects, the study is largely comprised of additional data from other studies. I struggled to understand why only one group of dogs performed one of the behavioral tests, and then how the genetic pieces from dogs not tested in that behavioral task, and the additional task, fit into the study presented here. Please develop the ideas fully, making the rationale for the current approach clear and being very explicit about how the work presented here expands upon and differs from work that you have previously published. Please reference the associated studies (e.g. lines 120-125). Please make clear what the behavioral variables are in the unsolvable task and how they were coded. Like Reviewer 1, I expect to see reliability analyses for all coded data.

I agree with the reviewers that data from the additional species (Shetland sheepdogs and border collies) should be included rather than excluded. If there is a lack of variability, this in itself may be of interest. However, if the lack of genetic variation made it impossible to analyze the results for these breeds, you might just delete reference to having collected that data. If you leave it in, you must include a reference for the study noted on line 145.

Several findings are overstated. For example, the ability of dogs to discriminate between human emotions is not the same as the ability to comprehend emotion (e.g., lines 69-70). Please note the authors of this research referred only to discrimination of emotions.

There are some additional references that could be included:

Gácsi, M., Vas, J., Topál, J., & Miklósi, Á. (2013). Wolves do not join the dance: Sophisticated aggression control by adjusting to human social signals in dogs.Applied Animal Behaviour Science, 145(3-4), 109-122. doi:http://dx.doi.org/10.1016/j.applanim.2013.02.007
Marshall-Pescini, S., Cafazzo, S., Virányi, Z., & Range, F. (2017). Integrating social ecology in explanations of wolf–dog behavioral differences. Current Opinion in Behavioral Sciences, 16, 80-86. doi:http://dx.doi.org/10.1016/j.cobeha.2017.05.002
Marshall-Pescini, S., Schwarz, J. F. L., Kostelnik, I., Virányi, Z., & Range, F. (2017). Importance of a species’ socioecology: Wolves outperform dogs in a conspecific cooperation task. PNAS Proceedings of the National Academy of Sciences of the United States of America, 114(44), 11793-11798. doi:http://dx.doi.org/10.1073/pnas.1709027114
Range, F., & Virányi, Z. (2013). Social learning from humans or conspecifics: Differences and similarities between wolves and dogs. Frontiers in Psychology, 4, 10. doi:http://dx.doi.org/10.3389/fpsyg.2013.00868
Range, F., & Virányi, Z. (2015). Tracking the evolutionary origins of dog-human cooperation: The “Canine cooperation hypothesis”. Frontiers in Psychology, 5, 2.
Werhahn, G., Virányi, Z., Barrera, G., Sommese, A., & Range, F. (2016). Wolves ( canis lupus) and dogs ( canis familiaris) differ in following human gaze into distant space but respond similar to their packmates’ gaze. Journal of Comparative Psychology, 130(3), 288-298. doi:http://dx.doi.org/10.1037/com0000036

I am surprised that you do not cite more of the work done by Friedericke Range

Please try to avoid paragraphs of one or two sentences (e.g., lines 80-83). Please fully develop paragraphs with introductory and summary statements, supported by facts.

Please do not use “since” or “while” unless in temporal context (e.g., line 58).

Reviewer 1 ·

Basic reporting

See below

Experimental design

See below

Validity of the findings

See below

Additional comments

This paper is a follow up to the same group’s interesting demonstration from a genome-wide association study that two SNPs from chromosome 26 are associated with human-directed social behavior in laboratory beagles (Persson et al., 2015).

The present paper extends those earlier findings by examining gene-behavior associations for the previously identified SNPs in two breeds of retrievers (Golden- and Labrador-retrievers). Genetic data is also presented for wolves; and mention is briefly made that border collies and Shetland sheepdogs were tested but results are not presented because “no significant results were found” (lines 352-353).

Overall I feel the paper has many interesting parts but they do not add up in the present draft to a satisfying whole. Sometimes what is presented comes across as off-cuts found on the lab’s cutting-room floor rather than a sustained piece of science.

In total five groups of canids receive some kind of consideration in this paper:
1. Wolves (from Swedish wildlife parks, no indication of the location of the original wild populations from which these animals were derived, not even the continent.) 19 individuals; genotyping; no behavioral work.
2. Golden retriever dogs (Swedish registered purebreds) 61 individuals; genotyping and behavioral testing on unsolvable problem and temperament test (not reported here). Half of the golden retrievers received an intranasal dosing with oxytocin prior to the behavior tests.
3. Labrador retrievers (Swedish registered purebreds) 100 individuals distinguished as 52 common-type and 48 field-type (i.e. bred for field work or not); genotyped and behavioral testing on unsolvable problem and pointing test.
4. Shetland sheepdogs; 28 individuals “tested and genotyped in same manner as the Labrador and golden retrievers” Data not presented.
5. Border collies 23 individuals “tested and genotyped in same manner as the Labrador and golden retrievers” Data not presented.

It is a great shame that no two types of canids are tested exactly equivalently (at least of the three types for whom results are presented – wolves, goldens and Labrador retrievers; the Shetlands and Border collies may have been tested identically to one of the breeds of retrievers, but their results are not presented beyond the comment: “the results do not contradict the findings in Labrador and golden retrievers” l. 351.) This makes it difficult to draw over-arching conclusions, and, indeed, the paper is rather thin on conclusions beyond generalities that the SNPs are associated with social behavior in the golden and Labrador retrievers – and alleles differ in frequency between the dog breeds and the wolves. They conclude that “This suggests that selection of favourable alleles in the genomic region of the SNPs has been an important part of dog domestication.” But I can’t see how they can conclude this. The genotype-behavior associations presented here appear to be different in the 2 breeds of dog for which fairly full results are reported – and even between the field and common Labrador retrievers. Consequently, the selection observed here could just as likely be due to selection in the creation of these breeds (likely less than 200 years ago) than in the original process of domestication (likely over 15,000 years ago).

The Figures only show significant findings which means that different behavior-genotype associations are presented for the golden and the Labrador retrievers. For the Goldens, associations of SNP1 with frequency of contact with the experimenter and frequency and duration of contact with the owner are presented. For Labrador retrievers and the same SNP1, associations of duration of owner gaze and mean pointing are presented. This means that a reader cannot affirm that the same trends of gene-behavior relationship are present in both dog breeds.

For a paper of this kind I would prefer to see all the gene-behavior relationships even those that are not significant. This would make it easier to conclude whether the patterns of relationship generalize across breeds or are different for the different breeds.

One other major point. In the Abstract and Introduction, the authors set up the argument that “During domestication, dogs have evolved human-directed social skills allowing humans and dogs to communicate and cooperate.” (Abstract) – and - “since during 15 000 years of parallel evolution, they have developed unique human analogue social skills (Hare & Tomasello 2005; Topal et al. 2009; Wang et al. 2016).” Lines 59-60.

I don’t think this line is defensible any more. Hare & Tomasello and Topal et al are now getting out of date as citations. Wang et al. 2016 does not present behavioral data.

The human-directed social skills cited have been demonstrated in numerous non-domesticated species and subspecies (e.g., Gacsi et al 2009; Giret et al., 2009; Hall et al., 2011; Pack & Herman, 2004; Range & Viranyi, 2011; Smet & Byrne, 2013; Schloegl et al., 2008; Scheumann & Call, 2004. Tomasello & Call, 2004; Udell et al. 2008; von Bayern & Emery, 2009). Thus they are not coupled to domestication. Insofar as there may exist human-direct social skills uniquely in dogs these may be the consequence of the unique environment in which pet dogs are reared, not anything about their genetics. Furthermore, insofar as there may exist genetically-influenced human-directed social skills uniquely in dogs, these may be due to recent selection in the formation of dog breeds within the last 200 years and not selection leading to domestication of wolves >15,000 years ago.

Overall I would like to see a revision of this paper that presents all the results from all the breeds, whether significant or not and which allows assessment of whether the observed genotype-behavior relations are possible evidence of a signal from domestication, or from the creation of these modern breeds of dogs. A paper like that could be very interesting to many readers.

Finally under major points – there is at least one other group addressing this issue in a related way: vonHoldt et al 2017 ought to be discussed here.

1. Minor points

Table 1. Gaze is not operationalized (it is defined as “gaze”). Needs more concrete operationalization of this behavior.

A sketch of unsolvable task needed

L 231-232. No choices need to be recorded somewhere; we cannot have trials that are lost from the analysis. And was the trial repeated until a choice was made? If so, the total number of trials needs to be included in the analysis.

Figure 3B. I don’t think the y axis scale is %.

References:
Giret, N., Miklósi, Á., Kreutzer, M., & Bovet, D. (2009). Use of experimenter-given cues by African gray parrots (Psittacus erithacus). Animal Cognition, 12(1), 1–10. https://doi.org/10.1007/s10071-008-0163-2
Gacsi, M., McGreevy, P., Kara, E., & Miklosi, A. (2009). Effects of selection for cooperation and attention in dogs. Behavioral and Brain Functions, 5(1), 31. https://doi.org/10.1186/1744-9081-5-31
Hall, N. J., Udell, M. A. R., Dorey, N. R., Walsh, A. L., & Wynne, C. D. L. (2011). Megachiropteran bats (Pteropus) utilize human referential stimuli to locate hidden food. Journal of Comparative Psychology, 125(3), 341–346.
Pack, A. A., & Herman, L. M. (2004). Bottlenosed dolphins (Tursiops truncatus) comprehend the referent of both static and dynamic human gazing and pointing in an object-choice task. Journal of Comparative Psychology, 118(2), 160.
Range, F., & Virányi, Z. (2011). Development of gaze following abilities in wolves (Canis lupus). PloS One, 6(2), e16888.
Scheumann, M., & Call, J. (2004). The use of experimenter-given cues by South African fur seals (Arctocephalus pusillus). Animal Cognition, 7(4), 224–230.
Schloegl, C., Kotrschal, K., & Bugnyar, T. (2008). Modifying the object-choice task: Is the way you look important for ravens? Behavioural Processes, 77(1), 61–65. https://doi.org/10.1016/j.beproc.2007.06.002
Smet, A. F., & Byrne, R. W. (2013). African Elephants Can Use Human Pointing Cues to Find Hidden Food. Current Biology, 23(20), 2033–2037. https://doi.org/10.1016/j.cub.2013.08.037
Tomasello, M., & Call, J. (2004). The role of humans in the cognitive development of apes revisited. Animal Cognition, 7(4), 213–215.
Udell, M. A. R., Dorey, N. R., & Wynne, C. D. L. (2008). Wolves outperform dogs in following human social cues. Animal Behaviour, 76, 1767–1773.
von Bayern, A. M., & Emery, N. J. (2009). Jackdaws respond to human attentional states and communicative cues in different contexts. Current Biology, 19(7), 602–606.
vonHoldt, B. M., Shuldiner, E., Koch, I. J., Kartzinel, R. Y., Hogan, A., Brubaker, L., … Udell, M. A. R. (2017). Structural variants in genes associated with human Williams-Beuren syndrome underlie stereotypical hypersociability in domestic dogs. Science Advances, 3(7), e1700398. https://doi.org/10.1126/sciadv.1700398

Reviewer 2 ·

Basic reporting

The authors did an excellent job securing a relatively large sample of golden retrievers and Labrador retrievers to examine whether there are associations between genotype and human-directed social behaviors in these breeds. Based on information provided in the Introduction and elsewhere, I am unclear on why the authors separated dog behaviors directed at the owner vs. experimenter and question whether it was necessary to analyze behaviors directed at owner and experimenter separately.

Overall, the study is well written.

The tables and graphs were well-done. However, since only behaviors associated with significant findings were presented in the graphs and the table associated with the models was only included as supplementary material, it was difficult to identify commonalities and differences between the two breeds studied.

Experimental design

The authors used solid experimental methods in their work with the two primary breeds that participated in the study; however, given that the golden retrievers and Labrador retrievers were recruited to participate in different studies, the presentation of findings is not as seamless as it could be (e.g. one breed of dogs received intranasal oxytocin and the others did not). I have provided suggestions below regarding ways to improve the presentation of findings regarding each breed. For example, I suggest including Table S1 in the actual paper rather than as a supplement. Also, more details are needed regarding how reliability of the behavioral coder(s) was assessed.

Lines 113-116: Perhaps the authors could add something in this section regarding how the wolf samples were ethically sourced? Did the procurement of the wolf DNA fall under the same permit as the one for the dog study (51-13)? Did anyone at the wildlife parks provide consent for the samples to be used?

Lines 131-132: What is meant by “wild captives or had been praised at the wildlife parks”? Were “wild captives” wild-caught? For the wolves that had been raised at the wildlife parks, for how many generations had their relatives been in captivity (and, presumably, engaging more with and relying more on humans than wild wolves)? Is it reasonable to assume that the captive wolves are representative of the wild wolf gene pool, or is there evidence of in-breeding or a severely limited gene pool in the captive population?

Lines 145-148: Based on the information provided in this paragraph, it seems it may not be worth further discussion of the Shetland sheepdogs or border collies in the Results or Discussion. I agree that it makes sense to share those findings as supplemental material, but if the authors felt the samples were too small to draw conclusions, I don’t think it is appropriate to make inferences about those breeds (e.g. “possible association in two additional breeds”—lines 383-384).

Lines 155-162: Given the authors’ previous findings of an association between OT and human-direct social behavior, and given the focus of the current study on human-directed social behavior, it seems the main effects of OT treatment and any interactions between OT treatment and genotype should be reported within the manuscript rather than as supplemental information.

Lines 208-210: Who coded the videos? Was inter-rater reliability established? I agree that the ethogram (Table 1) is necessary, but I think it would be good to mention within the body of the paper that the behaviors scored focused on proximity, physical contact, and gaze in relation to the experimenter and owner.

Lines 305-341: It appears that each of the six behaviors was tested as the dependent variable for SNP1 and SNP2, which means 12 models were examined—although only the statistically significant findings are reported in the text. Was any statistical correction made for running so many models?

Also, due to the number of models run and differences in significant findings for golden retrievers and Labrador retrievers, I find it challenging to identify ways in which findings from the two breeds complement each other and ways in which they diverge. It would be helpful if the authors could address the similarities and differences more explicitly in the Discussion.

Additionally, what are we to make of owner-oriented behaviors vs. experimenter-oriented behaviors? Did the authors hypothesize genotype would impact engagement with owners and experimenters differently? I don’t see any mention of this in the Introduction. If there were no hypotheses about whether dogs of particular genotypes would engage differently with owners vs. experimenters, might it make sense to combine engagement with owner and experimenter? This would reduce the number of models run, as there were be three dependent variables (gaze, proximity, and physical contact) rather than six.

Line 343: So that the reader can readily view all findings, significant and non-significant, I suggest including Table S1 within the actual paper rather than as a supplement.

Validity of the findings

I am unsure what to make of the findings. The authors ran six models for SNP1 and six for SNP2 for golden retrievers and for Labrador retrievers. The way the data are presented, it is difficult to identify similarities between breeds regarding the relationship between genotype and behavior, and I’d like to see the authors provide more of a synthesis in the Discussion regarding commonalities and differences between the findings for golden retrievers vs. Labrador retrievers.

Lines 348-354: In the Methods section, the authors mention data for border collies and Shetland sheepdogs was limited and that “they were only included as supplementary data” (line 148). Thus, I think this paragraph is unnecessary.

Lines 427-429: Why do the authors believe this behavioral difference is due to selection rather than differences in the types of experiences had by field vs. common type Labrador retrievers? In other words, might owners of field Labradors be more likely to encourage contact seeking than owners of common Labradors?

Lines 435-443: This does not read as a cohesive paragraph and is not a strong way to wrap up the Discussion. It would be better wrap up the Discussion by describing the study’s main strengths and limitations and clearly connect the value this research holds for understanding dog evolution as well as human social behavior and disorders.

Additional comments

Line 27: Change “associated to” to “associated with.”

Lines 45-47: Consider beginning the Results section of the abstract with these sentences about genetic variation (or lack of genetic variation) in the dog breeds and wolves tested.

Line 91: Consider rephrasing “social disorders in humans such as…” as “human social disorders such as…”

Lines 92-94: This sentence is vague and lacks clarity.

Line 176: I am unsure what is meant by “one and the same for each of the breeds.”

Line 177: Change “experimenter that” to “experimenter who.”

Line 231: Just double-checking that the dog only had 3 seconds to make a choice? This seems like a rather short time period, but then again, dogs can be quite fast.

Line 241: I am familiar with the term “whole blood” but not “full blood.” Are the authors referring to “whole blood”?

Table 1: Were “zone” and “physical contact” mutually exclusive behaviors? The way the zone behavior is defined, a dog could be in physical contact (within one body length) and simultaneously in physical contact.

---

## Round 0.2 · Minor Revisions

Fortunately, one of the previous reviewers was available to review your revision and was completely satisfied. Thank you for your careful attention to the reviewers’ comments in the previous round. I have read the revision carefully myself and I have a few remaining minor comments, and one more significant concern. Please address these remaining issues. Comments refer to the non-tracked PDF of the MS.
I apologize for not identifying this issue with the last round but, now with the increased focus of the revised MS, it is easier to attend to issues with the unsolvable task in particular. If you code for attention-seeking behavior toward the humans, but not for persistence in the task, it is possible that you find evidence of a link between SNPs and social behavior that is really due to differences in motivation or persistence. That is, maybe dogs are spending more time looking to humans because they are not persisting in solving the task, or because they are. I think it’s important to rule that out by also coding for the dog’s behavior toward the apparatus, not just their behavior to the owners and experimenters.

On the first line of the Abstract, Background “allows” should be “allow.”
The sentence on lines 61-62 is awkward. Please reword. It is not the talents that outperform another species. I think you should also be more cautious in drawing comparisons between wolves and dogs and dogs and chimpanzees as methodology across species often differs in important ways, and several studies directly comparing wolves and dogs have failed to replicate key differences in ability when methods are better controlled. Chimpanzees are kept in an unnatural setting when housed in captivity whereas for dogs, captive environments are typical, so I think that issue needs to be considered in testing situations.
I think the statement on line 68 is also overstated and requires more explanation.

On line 75, use “and” instead of & except when in parentheses.
There should be a comma after hypothesis on line 77.

On line 80, delete the “the” at the beginning of the sentence and change “have” to “has.”

The references on line 81 include some additional text that needs to be deleted.

There are still some instances of “since” being used in a non-temporal context (e.g., line 92). Please change to “because.”

Please place commas around “due to this” on line 148 and “even though….wolf” on lines 447-448.

Please state very clearly in the MS whether the data reported here have been reported elsewhere (e.g., data on the unsolvable task for the retrievers seems to have already been reported in Sundman et al.)

I’m not sure it makes sense to suddenly mention that the experimenter was blind with regard to the hormone treatment, before describing the treatment presented in another paper (line 188).

On line 443, please move the “only” to after the ”in”

On line 454, please place a ; after goals, and a , after for example on line 455.

Please place a , after study on line 461.

Reviewer 2 ·

Basic reporting

The concerns I had with the initial submission have been addressed, and I have no additional concerns.

Experimental design

The concerns I had with the initial submission have been addressed, and I have no additional concerns.

Validity of the findings

The concerns I had with the initial submission have been addressed, and I have no additional concerns.

Additional comments

The concerns I had with the initial submission have been addressed, and I have no additional concerns.

---

## Round 0.3 · Minor Revisions

Thank you for your edits to the MS. I still have some minor edits that need to be fixed before I can formally accept the MS. As PeerJ doesn't offer copy-editing, it's important to take care of these at this stage. These comments refer to the tracked changes word document.

Line 47. Delete the extra period after selection.
Line 64. This wording is still incorrect. I suggest changing "even outperform" to "surpass."
The sentence on lines 77-78 needs work. Wolves didn't "become" dogs. Refer here to the evolutionary changes and differences between wolves and dogs. Do not start a sentence with E.g. Always follow e.g. and i.e. with commas.
On lines 80-81, could you say "are as attentive to human and conspecific actions as dogs are?
On line 95, 479, 485 and 489 change "persistancy" to "persistance"
On line 168 insert "between" before "2008.."
On lines 219-221, delete "Parts of the" and change "has" to "have"
On line 299 change "analysis" to "analyses" and change "was" to "were."
On line 301, place commas around "if necessary"
On line 444, do you mean "except for" or "in addition to?"
On line 469, move the "only" to before "a very limited sample."
On line 488, place a comma after thus.
Break the sentence on lines 495-497 into two sentences. It is not grammatical as written.

---

## Round 0.4 · accepted · Accept

Thank you for attending to the final list of changes.

#